# Exploration of Filled-In Julia Sets Arising from Centered Polygonal Lacunary Functions

**L.K. Mork [1], Trenton Vogt [1], Keith Sullivan [1], Drew Rutherford [2] and Darin J. Ulness [2,*]**

1   Department of Mathematics, Concordia College, Moorhead, MN 56562, USA
2   Department of Chemistry, Concordia College, Moorhead, MN 56562, USA
*   Correspondence: ulnessd@cord.edu

**Abstract:** Centered polygonal lacunary functions are a particular type of lacunary function that exhibit properties which set them apart from other lacunary functions. Primarily, centered polygonal lacunary functions have true rotational symmetry. This rotational symmetry is visually seen in the corresponding Julia and Mandelbrot sets. The features and characteristics of these related Julia and Mandelbrot sets are discussed and the parameter space, made with a phase rotation and offset shift, is intricately explored. Also studied in this work is the iterative dynamical map, its characteristics and its fixed points.

**Keywords:** fractals; Julia sets; lacunary functions; Mandelbrot sets; centered polygonal numbers; iterative dynamics

---

## 1. Introduction

Analytic functions have long played an important role in physics. Isolated singularities of analytic functions often provide physical insight because they carry much of the information about the function itself. These isolated singularities restrict the radius of convergence of the power series associated with the analytic functions. Much of the early work on analytic functions was on developing methods to analytically continue functions outside the radius of convergence [1,2].

In certain cases, the singularities condense into a dense curve in the complex plane called a natural boundary. Analytic continuation is not possible through the natural boundary. Lacunary functions are a particularly important class of functions that exhibit a natural boundary [1,3,4]. These functions are also referred to as gap functions. Lacunary functions are characterized by a Taylor series that has "gaps" (or "lacunae") in the progression of powers. That is, only certain powers in the power series are active. An example of such a function is $f(z) = \sum_{n=1}^{\infty} z^{n^2} = z + z^4 + z^9 + z^{16} + \cdots$. This work is focused on the special family of lacunary functions that occur when the powers are given by the centered polygonal numbers. These are the monotonically increasing sequence of numbers associated with the points on a polygonal lattice [5,6]. The family is referred to as the centered polygonal lacunary functions.

Hadamard's gap theorem states that if the gaps in the powers increase such that the gap tends to infinity as $n \to \infty$, then the function will exhibit a natural boundary [4]. In the example function above, $f(z)$ is analytic in the open unit disk, with the natural boundary being the unit circle itself. The same is true for the centered polygonal lacunary functions.

The undesirable behavior of the lacunary functions have limited their use in application to physical problems. Nevertheless, lacunary functions have been investigated in some physical settings recently, where the presence of natural boundaries have been shown to have real physical consequences. Shado and Ikeda have demonstrated that natural boundaries impact quantum tunneling in some systems by influencing instanton orbiting [7]. Creagh and White showed that natural boundaries can be important in the short-wavelength approximation when calculating evanescent waves outside of

elliptic dielectrics [8]. Greene and Percival discussed natural boundaries in the context of Hamiltonian maps in the area of integrable/nonintegrable systems [9]. Guttmann et al. have shown that if an Ising-like model system is not solvable, then any solution must be expressible in terms of functions having natural boundaries [10,11]. Further, Nickel has explicitly shown the presence of a natural boundary in the calculation of the magnetic susceptibility in the 2D Ising model [12]. Recently, Yamada and Ikeda have investigated wavefunctions associated with Anderson-localized states in the Harper model in quantum mechanics [13]. In kinetic theory, lacunary functions exhibit features upon approaching the natural boundary that are related to Weiner (stochastic) processes. As such, lacunary functions have been discussed in the context of Brownian motion [4,14,15].

Along with these applications in physics, lacunary functions have found utility in probability theory. Certain lacunary trigonometric systems behave like independent random variables. Most notably, they are consistent with the central limit theorem [2,16–18]. Lacunary trigonometric systems are also of interest in the study of harmonic analysis on compact groups [19,20], Gaussian summations [21,22] and the Jacobi theta functions [23–25]. Notably connected to the current work Vartziotis and Bohnet have recently invested the fractal character of certain trigonometric series [26].

Very recently, the current authors have studied a particular family of lacunary functions called centered polygonal lacunary functions [27]. These are the lacunary functions where the active terms in the Taylor series are those whose powers are centered polygonal numbers. These lacunary functions (and their attendant finite sequences) have several unique properties that set them apart from general lacunary functions. Notably, they have true rotational symmetry. Germane to the current work is the observation that centered polygonal lacunary functions can exhibit fractal-like characteristics with self-similarity often arising from various points of view [3]. Costin and Huang have recently published an interesting investigation of lacunary functions near the naural boundary and have shown self-similar and fractal-like character in these systems. In the last couple of years, Nazeer and Kang have collaborated on developing fractal generating and escape time algorithms for finite polynomials based on the concept of S-convexity [28–31].

The purpose of the current work is to look more deeply into the fractal features of the centered polygonal lacunary functions through a thorough study of their associated Julia and Mandelbrot sets. In a related way, the lacunary functions can be viewed as a (infinite) discrete dynamical system. The orbits and fixed points of their systems have interesting properties.

Although all of the numerical calculations for this work were direct summation of the lacunary sequence members by MATHEMATICA, some recent work by Yamada and Ikeda has investigated the use of Padé approximate methods for speeding up the convergence of summations for lacunary functions [13,32].

*Main Result*

The main result of this work is the detailed look at the fractal character of the centered polygonal lacunary functions. This is exposed through analysis of the filled-in Julia sets via exploration of a parameter space that includes an iterative offset shift as well as a phase rotation. These filled-in Julia sets exhibit interesting qualitative and quantitative behavior that is examined in detail below. The fractal character of the centered polygonal lacunary functions is also exposed through analysis of its iterative dynamics. New and interesting fixed point behavior is seen as a function of the phase rotation parameter.

The motivation for this study comes from two distinct points-of-view. First, centered polygonal lacunary functions have not been significantly studied and they have a number of unique features. Their fractal character is one of those features. Second, it is hoped that the family of centered polygonal lacunary functions can serve as a source of insight into Julia sets, Mandelbrot sets and iterative dynamics in their own right.

## 2. Centered Polygonal Lacunary Functions

Throughout this work, a *lacunary sequence* of functions, with the $N^{\text{th}}$ member of the sequence is given by the partial summation

$$f_N(z) = \sum_{n=1}^{N} z^{g(n)}, \tag{1}$$

where $g(n)$ is a function of $n$ satisfying the conditions of Hadamard's Gap Theorem, which we list here directly from Reference [4] for the convenience of the reader.

**Theorem 1.** *The function $f_\infty(z)$ from Equation (1) has a natural boundary to its circle of convergence if there exists a fixed $\Lambda > 1$ such that for all n*

$$\frac{g(n+1)}{(g(n))} \geq \Lambda. \tag{2}$$

Note that while the summation in Equation (1) could be defined to start at $n = 0$, for this work, it is more desirable to it start at $n = 1$.

For notational convenience we define

$$\mathfrak{L}(g;z) \equiv \left\{ f_N(z) = \sum_{n=1}^{N} z^{g(n)} \right\}, \tag{3}$$

to represent the particular lacunary sequence described by $g(n)$. Two examples which illustrate the notation are as follows:

- $\mathfrak{L}(n^2;z) \equiv \left\{ f_N(z) = \sum_{n=1}^{N} z^{n^2} \right\}$
- $\mathfrak{L}(n^2;\rho e^{i\pi/3}) \equiv \left\{ f_N(z) = \sum_{n=1}^{N} (\rho e^{i\pi/3})^{n^2} \right\}$.

These partial summations will truncate after the $g(N)^{\text{th}}$ power and have $N$ terms in them. This is true for all lacunary functions but this work is only focused on centered polygonal lacunary functions. Centered polygonal numbers and their properties were explored and are discussed in the next section.

### 2.1. Centered Polygonal Numbers

Centered polygonal numbers (cpn) are an infinite, increasing sequence of numbers associated with points on a polygonal lattice [5]. These numbers were discussed in the context of two-dimensional crystal structures by Teo and Sloane [6]. The formula for the centered $k$-gonal numbers where $n \in \mathbb{N}^+$ is

$$C^{(k)} = \left\{ \frac{kn^2 - kn + 2}{2} \right\}. \tag{4}$$

when it is necessary to identify a particular member of $C^{(k)}$, the notation $C^{(k)}(m)$ will be used for the $m^{\text{th}}$ member of the set.

These centered polygonal numbers carry an interesting property, in that any number of the form $kn + 1$ can be written as the sum of as the sum of $k + 1$ members of $C(k)$. Gauss proved a similar property using the triangular numbers.

Gauss's "Eureka Theorem" states that every integer, $n$, can be written as the sum of three triangle numbers.

$$n = T(n_1) + T(n_2) + T(n_3). \tag{5}$$

**Theorem 2.** *Every number of the form $kn + 1$, where $k, n \in \mathbb{N}$ can be written as a sum of $k + 1$ members of $C^{(k)}$.*

**Proof.** Notice first, 1 is a member of $C^{(k)}$, so this is equivalent to stating that every number of the form $kn$ can be expressed as the sum of $k$ members of $C^{(k)}$. Now, the sum of $k$ such members has the form

$$\frac{k\,n_1^2 - k\,n_1 + 2}{2} + \frac{k\,n_2^2 - k\,n_2 + 2}{2} + ... + \frac{k\,n_k^2 - k\,n_k + 2}{2} \tag{6}$$

where $n_i \in \mathbb{N}^+$ with $i \in I$. This would occur if and only if

$$\frac{k(n_1^2 + n_2^2 + ... + n_k^2) - k(n_1 + n_2 + ... + n_k) + 2k}{2} \in k\mathbb{N}$$

$$\iff k(n_1^2 + n_2^2 + ... + n_k^2) - k(n_1 + n_2 + ... + n_k) + 2k \in 2k\mathbb{N}$$

$$\iff (n_1^2 + n_2^2 + ... + n_k^2) - (n_1 + n_2 + ... + n_k) \in 2\mathbb{N}$$

$$\iff n_1(n_1 - 1) + n_2(n_2 - 1) + ... + n_k(n_k - 1) \in 2\mathbb{N}$$

After a change of index,

$$\iff \frac{(n_1 + 1)(n_1)}{2} + \frac{(n_2 + 1)(n_2)}{2} + ... + \frac{(n_k + 1)(n_k)}{2} \in \mathbb{N}$$

The final line shows that the original expression can, in fact, be written as a sum of at least three triangle numbers. This invokes Gauss's Theorem from above, completing the proof. $\square$

To give some general information on the nature of centered polygonal lacunary functions, a few observations are stated below. These observations are all based on the set of numbers $C^{(k)}$ and, more specifically, the set generated by taking all possible non-zero, positive differences between these numbers: $\Delta C^{(k)} \equiv \left\{ C^{(k)}(p) - C^{(k)}(q) \right\}$, $p > q$. (There is, of course, a similar set where all differences are negative). The following observations are noted [27]:

- All members of $\Delta C^{(k)}$ are divisible by $k$.
- The difference between the $(m + n)^{\text{th}}$ and the $n^{\text{th}}$ term is

$$C^{(k)}(m + n) - C^{(k)}(n) = k\left( mn + \frac{m(m - 1)}{2} \right) \tag{7}$$

- The "normalized" set $\Delta C = \frac{\Delta C^{(k)}}{k}$ is independent of $k$.
- All integers are represented in $\frac{\Delta C^{(k)}}{k}$. Although, for any truncated set $C^{(k)}$ having a finite $P$ number of members, there are missing integers. The first of which is $2^m$ where $m$ is the value such that $2^m > P$.
- Some integers are represented more than once.

*2.2. Features of Lacunary Sequences*

The most revealing way to interpret members of $\mathfrak{L}(g; z)$ is in their polar form: $f_N(z) = R(z)e^{i\Phi(z)}$ and to focus on $R(z) = |f_N(z)|$, although $f_N(z)$ itself is also discussed. Figure 1 shows this representation. Specifically, one can use the example of $\mathfrak{L}(C^{(4)}; z)$ (bottom left of Figure 1). The contour plot is of $|f_{16}(z)|$, however the plot has been limited from $|f_{16}(z)| = 0$ to $|f_{16}(z)| = 1$. The $|f_N(z)| = 1$ is called the unity level set.

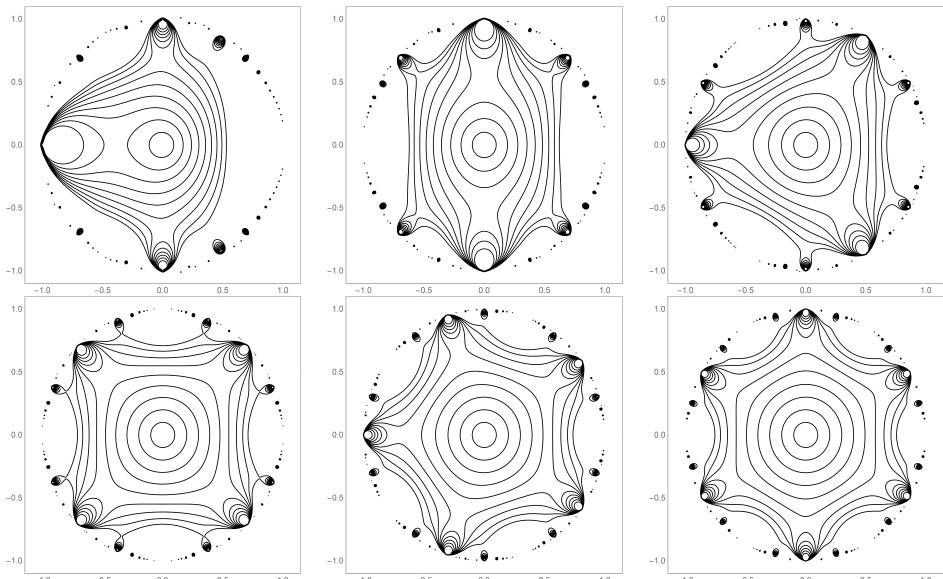

**Figure 1.** Contour plots of $f_{16}^{(k)}$ for value of $k$ 1–6. The contour plots are truncated at the unity level set ($f_{16}^{(k)}(z) = 1$). The most prominent feature of a given graph constitutes a subset $f_{16}^{(k)} < 1$, which is path-connected to the origin. The complicated nature of $\mathfrak{f}_{16}^{(k)}$ produces a host of additional subsets, $f_{16}^{(k)} < 1$, that are not path-connected to the origin. These appear adjacent to the natural boundary.

An important property is exposed when a cpn-based lacunary function is graphed. The cpn-based functions display a perfect *k*-fold rotational symmetry (as demonstrated in Figure 1). The primary symmetry will manifest itself in powerful ways in the associated Julia ns Mandelbrot set, as discussed at length in the sections below. The driver at this *k*-fold symmetry is a property of the cpn-based function called primary symmetry, which is defined in general as follows.

**Definition 1.** *Primary symmetry. The rotational symmetry of the $N = 2$ member of $|\mathfrak{L}(g; z)|$, $|f_2(z)|$, is called the* primary symmetry.

**Theorem 3.** *The primary symmetry of $|\mathfrak{L}(g(n); z)|$ is $K = g(2) - g(1)$.*

**Proof.** Casting $z$ as $\rho e^{i\phi}$ results in

$$
\begin{aligned}
|f_2(z)| &= \sqrt{\left( (\rho e^{-i\phi})^{g(1)} + (\rho e^{-i\phi})^{g(2)} \right) \left( (\rho e^{i\phi})^{g(1)} + (\rho e^{i\phi})^{g(2)} \right)} \\
&= \sqrt{\rho^{2g(1)} + \rho^{2g(2)} + \rho^{g(2)+g(1)} \left( e^{i(g(2)-g(1))\phi} + e^{-i(g(2)-g(1))\phi} \right)} \\
&= \sqrt{\rho^{2g(1)} + \rho^{2g(2)} + 2\rho^{g(2)+g(1)} \cos((g(2) - g(1))\phi)}.
\end{aligned}
\tag{8}
$$

This implies $|f_2(z)|$ has primary symmetry of $k = g(2) - g(1)$, completing the proof. $\square$

In general, the higher order terms end up breaking rotational symmetry. For example, $\mathfrak{L}(n^2; z)$ will exhibit $2^2 - 1^2 = 3$-fold quasi-symmetry [27] but not true 3-fold rotational symmetry. It so happens that $\mathfrak{L}(C^{(k)}; z)$ continues to hold true rotational symmetry for all members of the lacunary sequences; a rare occurrence. For cpn-based lacunary functions, $K = C^k(2) - C^k(1) = K$. Since centered polygonal lacunary functions are exclusively used in this work, $f_N(z)$ is further decorated as $f_N^{(k)}(z)$ to capture the primary symmetry as well as the $k$ value for the function.

The lacunary sequence $|\mathfrak{L}(C^{(4)}; z)|$ again can be used as a specific, illustrative example of primary symmetry. One calculates the primary symmetry to be $k = 4$. This is clearly seen in lower left box of Figure 1.

A few more observations are in order. Common features of cpn-based lacunary functions include mirror symmetry about the real number line and the emergence of the natural boundary structure on the unit circle. The mirror symmetry between the upper and lower-half planes is simply a consequence of the general properties of the modulus of a function. The natural boundary is not simply a continuous curve of singularity at the unit circle but rather a dense arrangement of both singularities and zeros. The plots in Figure 1 display some of the emerging zeros. If the sums are taken out to larger $N$ values, more of these features appear along the unit circle.

## 3. Iteration of Centered Polygonal Lacunary Functions

Lacunary sequences themselves have been studied previously [27], so this work will focus primarily on the related Julia and Mandelbrot sets. The construction of a Julia set, of course, requires iteration. The following notation is used to capture the number of iterations ($j$), the order of the partial summation ($N$) and the primary symmetry ($k$).

$$
{}^{j}h_N^{(k)}(z) \equiv f_N^{(k)}(\overbrace{f_N^{(k)}(f_N^{(k)}(...f_N^{(k)}(z))))}^{j}, \tag{9}
$$

where $f_N^{(k)}(z)$ iterated $j$ number of times. And $N$ is the number of terms in the partial summation.

*Coefficients of the Iterated Centered Polygonal Lacunary Function*

It is clear that iteration will result in a new power series with different acting monominal powers and coefficients that are no longer simply connected. Although the powers may be simple and calculated easily using theorem 2, the coefficients will increase with each iteration. This increase was studied to observe the possibility of a correlation between the coefficients and the $k$ value as well as $j$ number of iterations. The coefficients are denoted by $a_n$, where $n$ is the same $n$ as in $C^{(k)} = \left\{ \frac{kn^2 - kn + 2}{2} \right\}$. Algorithms for the coefficients of the first 4 terms are given below.

$$
a_1 = 1
$$
$$
a_2 = j
$$
$$
a_3 = \frac{(k+1)(j^2+j)}{2}
$$
$$
a_4 = \left( \frac{5}{12}k^2 + \frac{3}{4}k + \frac{4}{2} \right) j - \left( \frac{3}{4}k^2 + \frac{5}{4}k + \frac{1}{2} \right) j^2 + \left( \frac{1}{3}k^2 + \frac{1}{2}k + \frac{1}{6} \right) j^3
$$

Beyond this, the sequence of coefficients rapidly becomes complicated. However, there are some notable trends. The $a_n$ values are observed to be $(n-1)^{th}$ degree polynomial with respect to $j$ for $n \geq 1$. Additionally, the coefficient for the $j$ values are observed to be $(n-2)^{th}$ degree polynomials, with respect to $k$ for $n \geq 2$.

## 4. Family of Base Julia Sets and the Mandelbrot Sets

It is typical to add an offset constant to $z$ each iteration with the textbook example being $f(z) = z^2 + c$. Each $c$ gives rise to a Julia set. This mechanism is explored in this work for the case of centered polygonal lacunary functions. In addition, however, a second parameter is included: a multiplicative phase factor. The iterated function is then ${}^{j}h_N^{(k)}(e^{i\theta}z + c)$. A discussion of the methods used throughout the is work is given in Section 8.

Exploration of the Julia sets associated with points in parameter space $(\theta, c)$ is the content of this and the next few sections. The Julia sets that arise from the origin of the parameter subspace $(0,0)$ are referred to as the base Julia sets. Those associated with the parameter subspace $(\theta, 0)$ are called the *phase rotated Julia sets* and those associated with the parameter subspace $(0, c)$ are called the *shifted*

*Julia sets*. By focusing on these two subspaces, a significant amount of information about the whole parameter space can be obtained. For the purpose of this paper, the filled-in Julia set is used to show these subspaces, as it gives insight into some of the properties of the Julia set itself.

As seen in Figure 2, each filled-in Julia set displays perfect $k$-fold symmetry, much like the corresponding contour plots in Figure 1. The filled-in Julia sets also display similarly shaped lobes for $k$ values greater than one, with the number of lobes being equal to $k$. Between these lobes, there is a penetrating cleft structure that is directed towards the origin of the filled-in Julia set. This structure penetrates deeper into the filled-in Julia set as $j \rightarrow \infty$. The rate at which this happens can be shown by finding the radius of the largest inscribed circle at a given $j$ value and comparing it to the radii of other $j$ values (as seen later).

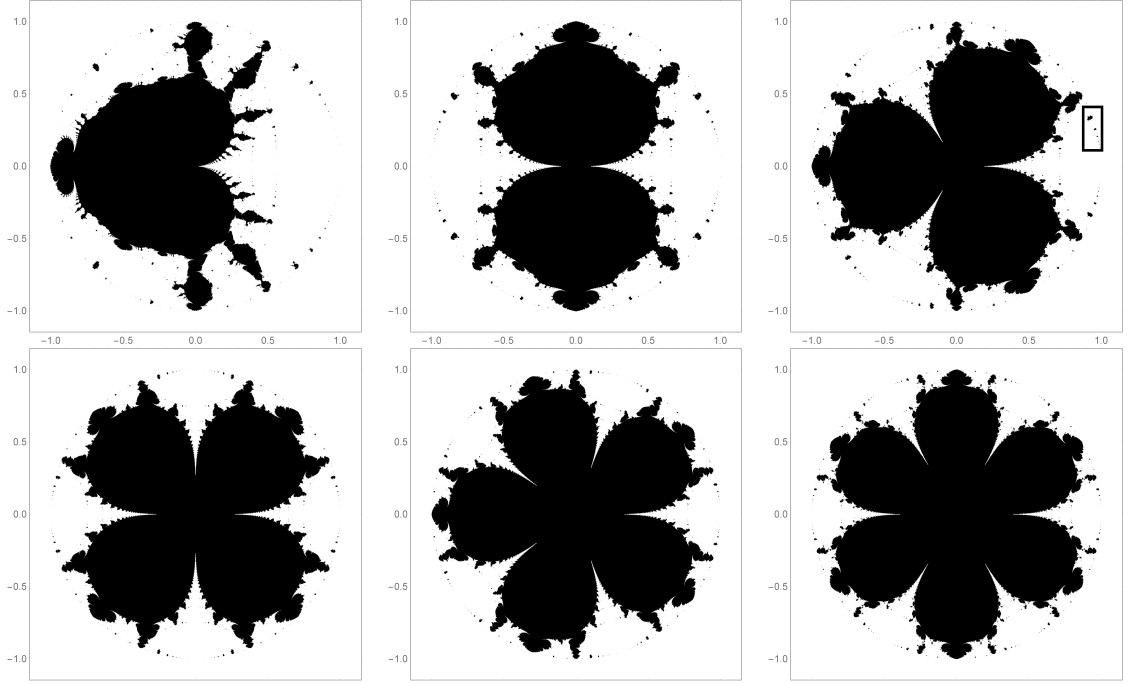

**Figure 2.** Base filled-in Julia sets created from the centered polygonal lacunary functions ${}^{50}h_{30}^{(1)}(z)$ through ${}^{50}h_{30}^{(6)}(z)$. These filled-in Julia sets arise from the corresponding lacunary functions seen in Figure 1 for $\theta = 0$ and $c = 0$. Notably, they exhibit the same $k$-fold symmetry as their corresponding lacunary functions. The points in black represents those points that converge to zero, whereas the white points diverge towards infinity. The principle subset of the filled-in Julia set contains the origin. The subsets not path-connected to the origin are called "islands" (when large) and "islets" (when small). A chain of islands or islets is referred to as an archipelago. (For example, the archipelago in the black box on the upper right graph).

As is commonplace when describing fractals, it is valuable to use non-technical descriptive terms for various features of the fractal. Clearly noticeable from Figure 2 is the fact that the filled-in Julia set contains subsets that are not path connected to the principle subset (that which contains the origin). These subsets will be referred to subsequently as "islands" (when large) or "islets" (when small). Chains or clusters of islands/islets will be referred to as an "archipelago" (see, for example, the set of islets contained in the black box which is superimposed on the upper right panel in Figure 2).

A striking feature seen in the filled-in Julia sets, like those shown in Figure 2, are the archipelagos that extend along the edge of the natural boundary. More archipelagos systematically connect like-features across the penetrating cleft.

A Mandelbrot set is created in the standard way by offsetting the base Julia set of the the corresponding $k$-value. This is done by changing the $c$ value, which is a complex number, in the

equation $^{j}h_{N}^{(k)}(z+c)$. The shifted Julia sets arising from a given $c$ value will be described in more detail in a later section.

As seen in Figure 3, each of the Mandelbrot sets show a symmetry about the real axis. The Mandelbrot sets also display a rotational array of lobes with the angles in-between being equal to that of $\frac{360°}{k}$. Note the areas of diverging points that penetrate towards the origin of the Mandelbrot set occurring between lobes, much like the penetrating cleft in the Julia set. Each individual lobe becomes increasingly smaller in area as $k$ increases. Larger island structures appear along radial angles of multiples of $\frac{360°}{k}$. Archipelagos string between adjacent lobes.

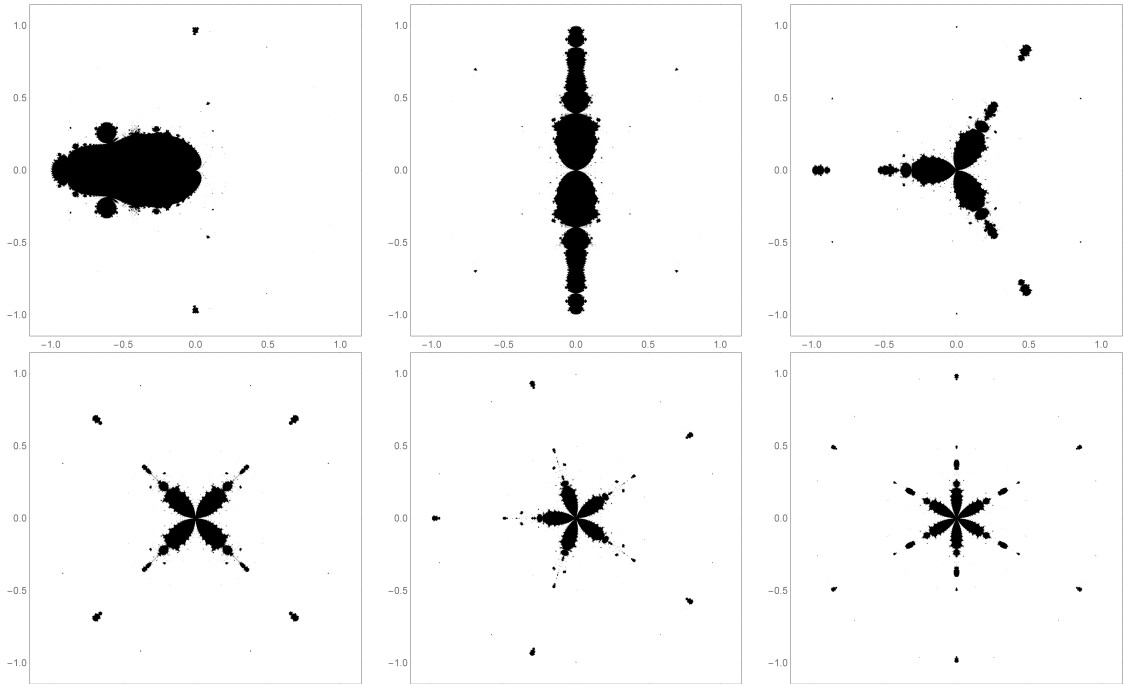

**Figure 3.** Mandelbrot sets associated with the corresponding lacunary functions (Figure 1 and Julia sets (Figure 2) $^{50}h_{30}^{(1)}(z)$ through $^{50}h_{30}^{(6)}(z)$. From top left to bottom right the Hausdorff modified dimension (described in detail in a later section) of the Mandelbrot sets are 1.363, 1.302, 1.300, 1.335, 1.307, 1.261, 1.321 and 1.247 respectively.

## 5. Julia Sets for Iterative Phase Rotations

### 5.1. Whole Julia Set Phase Shift

Iterative phase rotation is quite a productive area when it comes to Julia sets. Phase rotations occur when the $c = 0$ in $^{j}h_{N}^{(k)}(e^{i\theta}z+c)$ and only the $\theta$ value is changed. Each filled-in Julia set with a $\frac{360°}{k}$ phase rotation ($\theta = \frac{360°}{k}$) will be identical to the filled-in Julia set with no rotation for the respective $k$. This is useful computationally, because one need only calculate $\left(\frac{1}{k}\right)^{th}$ of a full circle. The example of $k = 3$ is shown in Figure 4. Many features of this case are common to the case at an arbitrary $k$. First, the filled-in Julia set for a small phase shift (5° shown in the upper right graph of Figure 4) has distinct spirals that curl inward. The depth of these curls is increasing with increasing $j$ value. Second, no matter the phase rotation or the $k$ value, the Julia set retains its $k$-fold rotational symmetry (three-fold rotational symmetry in the case of Figure 4). A final observation is the presence of the previously discussed archipelagos, regardless of $\theta$ value.

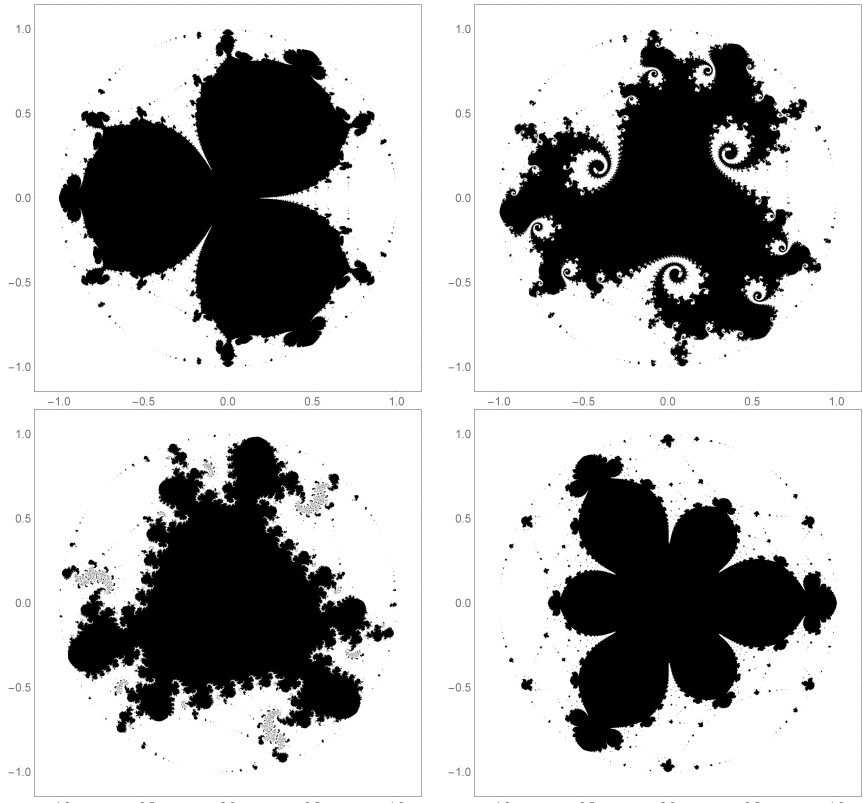

**Figure 4.** The filled-in Julia set for $^{50}h_{30}^{(3)}(z)$. The top left shows the filled-in Julia set for $^{50}h_{30}^{(3)}(z)$. The top right shows that same filled-in Julia set with a 5° phase rotation, $^{50}h_{30}^{(3)}(z)$. The bottom left shows the Julia set with a 19.5° phase rotation. And finally the bottom right shows the Julia set with a 60° phase rotation. Note that phase rotation breaks mirror symmetry but maintains *k*-fold rotational symmetry.

Another item of note is that when there is not a phase rotation, the distance from the origin to the nearest divergent point is the smallest it will be (see left image of Figure 5). The next smallest distance is when the $\theta$ value is exactly $\frac{360°}{2k}$. This graph is shown for $k = 3$ in the bottom right of Figure 4 and the values of radius are shown in the middle graph of Figure 5. This property of nearest divergent point is also illustrated in Figure 5 as a function of *j*. The left plot of Figure 5 shows the largest radius that can be made centered at the origin when the number of iterations (*j*) of the function is increased. As is shown, when *j* is relatively small, the penetration of the cleft is relatively shallow. However, as *j* increases, the radius of a circle centered at the origin can be fit to an exponential decay (seen in the left graph of Figure 5).

The center graph of Figure 5 shows the radius of the largest circle that can be made centered at the origin as a function of $\theta$. As can clearly be seen, the radius of the base Julia set is small but quickly increases as $\theta$ moves away from 0°. Then, the radius will continue to increase at a slower rate, with periodic sharp decreases. These are when the spirals of diverging points will uncurl towards the center. All the radii seem to be very nearly symmetrical about the $\theta = 60°$. This symmetry is also seen for the right graph of Figure 5. This is showing the area of non-diverging points as a percent of the unit disk. The symmetry about the mid point is shown, as well as a increase in area as $\theta$ approaches 60°.

An interesting phenomenon is exposed by comparing the bottom left graph to any of the others in Figure 4. By the right edge of any lobe of the 19.65° phase rotation graph, one can see a structure that is nearly severed from the primary domain. This structure does not exist in the top right graph but in the bottom left graph it can be seen that there are many disjointed pieces forming a general island. This forming occurs at about 19.65° and the dissolution was observed to occur at about 25°.

Figure 6 shows this region of the filled-in Julia set in fine detail. Two qualitatively different types of islands appear in this region. Some remain very similar across all three graphs. These are called stable islands. The second type changes dramatically across the three graphs. This is called an unstable island.

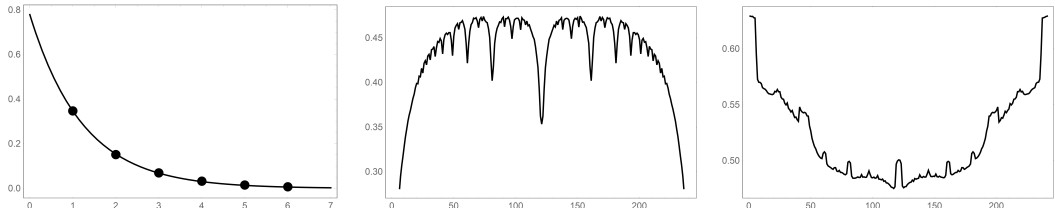

**Figure 5.** Radius of the Julia set for $f_4(z)$ of the sequence $^jh_4^{(3)}(z)$ versus $j$ is shown on the left. The radii are plotted along the ordinate and the abscissa is x, where $j = 10^x$. The data are fit to an exponential decay $Ae^{-\kappa x}$ (fit values: $A = 0.779$, $\kappa = -0.811$). These data expose the very slow ($\approx \log(\log(j))$) penetration of the cleft structures in the base filled-in Julia sets ($\theta = 0$, $c = 0$). Plots of radius, in the middle and the area as a percentage of the unit disk, on the right, of the Julia sets, as $\theta$ ranges from $0°$ to $120°$. The sharp downward peaks of the middle graph correspond to $\theta$ values where the spirals uncurl to stretch toward the origin. The area graph on the left shows that, although the graphs are not mirrored about the midpoint, the area of these graphs are similar.

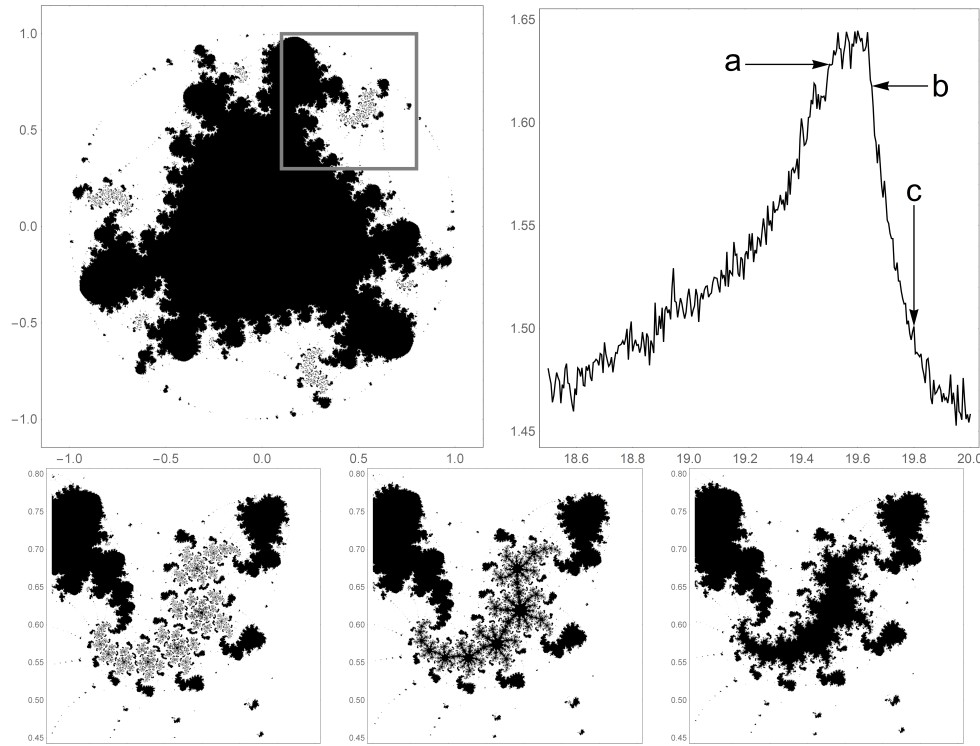

**Figure 6.** On the top left is shown the full filled-in Julia set for centered polygonal lacunary function $^{50}h_{30}^{(3)}(e^{i19.5°}z)$ with several islands shown in the box. On the top right is shown a graph of the dimension of the region in the box as the phase is ranging from $18.5°$ to $20°$. On the bottom left is shown island in the box with a $19.5°$ denoted a, phase rotation. In the center of the bottom is shown that same island with a $19.65°$, denoted b, phase rotation. On the bottom right is shown the island with a $19.8°$, denoted c, phase rotation. These bottom images have their dimension shown on the top right graph.

The bottom three graphs of Figure 6 show the unstable island before it connects, as it is connecting and after it has become one structure, respectively. Looking at the bottom left image, one can see

many self-similar structures make up this general island. This is intriguing as the structures in the middle of the unstable island mold together and change shape. Whereas, the stable islands and the one connecting the unstable island to the primary domain, don't change significantly through this $\theta$ range. The center image shows that there are many nexus that connect before the structure is complete (bottom right of Figure 6). The unstable island will fill out more, then at around $\theta = 25°$, break apart into many smaller self-similar structures.

### 5.2. Dimension of the Island and Julia Sets

When finding dimensions of fractal structures, two main methods are commonly used. The box-counting dimension simply creates a lattice grid over the fractal and counts the minimum number of boxes that intersect the fractal. Then, changing the side length of the boxes that make up the lattice grid before again counting the number of boxes which the fractal is intersected by [33]. In comparison, the Hausdorff dimension uses multiple circles of a small radius to cover a curve and determine the maximum number of overlapping circles of that radius that have nonempty intersections [33]. While the Hausdorff dimension has been proven to be more accurate when compared to fractals of the known dimension, the box-counting dimension still found a use in many cases.

For the purposes of this work, a modified version of the Hausdorff dimension was developed and employed (cf, Section 8). The modified Hausdorff dimension used is quite different from the standard method. Instead of using overlapping circles, the modified Hausdorff dimension made a set number of circles of varying sizes with the same center point. Within each circle, the number of members in the Julia set was counted. The logarithm of the radius of the circle was plotted against the logarithm of the number of members in the Julia set. With this modified Hausdorff dimension, one could find the overall dimension of the fractal as well as local dimensions.

Using this modified Hausdorff dimension gave dimensions very similar to that of fractals of known dimension. Though it has proven to be more accurate than the box dimension, the modified Hausdorff dimension took a long time to calculate, about 4–5 h for a single dimension calculation. Comparatively, the box-counting method took only about 50 seconds to calculate a dimension and it was still closely accurate when used on fractals of known dimension. Both methods are employed in this work and which particular one is used is indicated alongside the associated dimension data.

Now attention will be given to the dimension of the unstable island as phase rotation is changing. The top right plot of Figure 6 shows a graph of the dimension of the structure as the phase runs from $19.5°$ to $23.5°$. This covers the full range, from a not-connected structure, to a well established structure. This is seen from the points labeled a, b and c, which correspond to the bottom left, center and right images respectively. There is a large increase in dimension as this structure comes together and then the dimension follows a sharp decline as the island becomes a complete whole.

Figure 7 shows the dimension for the whole range of $\theta$ values for $^{50}h_{30}^{(3)}(e^{i\theta}z)$, when $\theta$ ranges from $0°$ to $120°$. This was done using the modified Hausdorff dimension. The dimension is far from constant as the set changes but the symmetry about $\frac{360°}{2k}$ can be seen. This is illuminating as the graphs are not similar about this center point but their dimensions are mirrored about $\frac{360°}{2k}$. There are also several angles where the dimension is much lower, primarily as $\theta$ approaches $0°$, $\frac{360°}{2k}$ and $\frac{360°}{k}$. These $\theta$ values are when the spirals uncurl to stretch toward the origin of the Julia set.

There are four ranges that have a significantly higher dimension then those surrounding them. These correspond to the Julia sets when the unstable islands, shown in Figure 6, are either coming together or breaking apart. This is believed to be due to the complexity of the many disjoint parts that form the island as well as the rest of the Julia set. It is noted that the overall range in dimension is similar to that for the canonical $f(z) = z^2 + c$ based fractals.

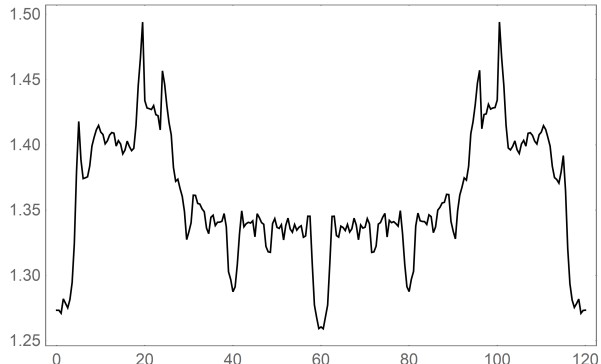

**Figure 7.** The plot of dimension for $^{50}h_{30}^{(3)}h(e^{i\theta}z)$ versus phase rotation $0°$ to $120°$. This was calculated using the modified Hausdorff dimension procedure.

### 5.3. Analysis of the Archipelagos as a Function of Phase Rotation

Some specifically interesting structures that arise when $^{50}h_{30}^{(k)}h(z)$ is graphed are the aforementioned islets (as pictured in a superimposed black box in Figure 2). These islets decrease in size as they approach the real axis, however, they remain mostly the same overall structure with $k$ number of lobes. The area ratio of these islets (used later in Tables 1 and 2) describes the size of one islet relative to the islet directly after it. As a numerical system, the islet assigned as the first islet was the islet closest to the first lobe (rotating counter-clockwise from the positive real axis). The subsequent islets are assigned increasing values as they approach the real axis. To find the dimension of each of the islets, the modified Hausdorff dimension was applied.

From Tables 1 and 2 above, one can see that the ratios from one islet to the next remains almost unchanged as the $k$ value increases, with an average standard deviation of 0.08205 and 0.02027 respectively. This shows that the relative size between islets is independent of $k$. When comparing the two tables, one also sees that this trend is also present when comparing the islets at a different $\theta$ value. This shows that the islets' relative size is also independent of the phase shift. Note that at $\theta = 0°$ there are only five islets and at $\theta = \frac{15°}{k}$ there are six islets. This accounts for some the differences seen in Table 1 as the islet that becomes the first islet for $k = 1$ simply doesn't detached from the main structure until a larger $\theta$ value. Also note that the ratios between the islets seems to decrease as one gets closer to the real axis, until the ratio from islet 4 to islet 5, where the ratio becomes larger. Then in Table 2, islets 5 and 6 are approximately the same size.

Tables 3 and 4 show that even with the islets decreasing in size, they differ very little in their dimension. This helps to show that the islets are actually a very similar shape to each other, not just qualitatively but also quantitatively. This is not to say that an islet's structure is independent of $k$. In reality, an islet's structure is unique to its $k$ value, containing $k$ number of lobes. Even with this unique structure, islets of the same number display a similar dimension to those of different $k$ values.

**Table 1.** The ratio of an islet's area to an identified islet's area for multiple $k$ values when $\theta = 0°$ and $c = 0$. For $k = 1$, only four islets are present, whereas for the other $k$ values there are five islets present. The table had an average standard deviation of 0.08205 between the data. This average standard deviation was found by averaging the standard deviation found for each row of data.

| | | | Islet Area Ratios $\theta = 0°$ | | | | |
|---|---|---|---|---|---|---|---|
| | $k = 1$ | $k = 2$ | $k = 3$ | $k = 4$ | $k = 5$ | $k = 6$ | $k = 10$ |
| 1 to 2 | 1.411 | 1.772 | 1.811 | 1.821 | 1.810 | 1.761 | 1.725 |
| 2 to 3 | 1.326 | 1.378 | 1.386 | 1.383 | 1.373 | 1.366 | 1.362 |
| 3 to 4 | 1.495 | 1.322 | 1.326 | 1.320 | 1.315 | 1.311 | 1.312 |
| 4 to 5 | | 1.507 | 1.511 | 1.508 | 1.504 | 1.810 | 1.503 |

**Table 2.** The ratio of an islet's area to an identified islet's area for multiple $k$ values when $\theta = \frac{15°}{k}$ and $c = 0$. The table had an average standard deviation of 0.02027 between the data. This average standard deviation was found by averaging the standard deviation found for each row of data.

| Islet Area Ratios $\theta = \frac{15°}{k}$ | | | | | | | |
|---|---|---|---|---|---|---|---|
| | $k = 1$ | $k = 2$ | $k = 3$ | $k = 4$ | $k = 5$ | $k = 6$ | $k = 10$ |
| 1 to 2 | 1.780 | 1.787 | 1.788 | 1.815 | 1.795 | 1.767 | 1.725 |
| 2 to 3 | 1.350 | 1.375 | 1.381 | 1.380 | 1.374 | 1.367 | 1.361 |
| 3 to 4 | 1.280 | 1.314 | 1.321 | 1.317 | 1.315 | 1.306 | 1.311 |
| 4 to 5 | 1.441 | 1.499 | 1.514 | 1.481 | 1.504 | 1.502 | 1.502 |
| 5 to 6 | 0.925 | 0.988 | 0.994 | 1.020 | 0.999 | 0.958 | 1.000 |

**Table 3.** The dimension of each of the islets when $\theta = 0°$ and $c = 0$. Again, $k = 1$ only had four islets while the rest of the $k$ values had 5 islets. The modified Hausdorff dimension was used to find the dimension of these islets.

| Islet Dimensions $\theta = 0°$ | | | | | | | |
|---|---|---|---|---|---|---|---|
| | $k = 1$ | $k = 2$ | $k = 3$ | $k = 4$ | $k = 5$ | $k = 6$ | $k = 10$ |
| Julia set | 1.327 | 1.268 | 1.277 | 1.254 | 1.277 | 1.321 | 1.276 |
| 1 | 1.350 | 1.297 | 1.316 | 1.278 | 1.296 | 1.343 | 1.283 |
| 2 | 1.345 | 1.300 | 1.314 | 1.274 | 1.297 | 1.323 | 1.310 |
| 3 | 1.328 | 1.298 | 1.283 | 1.263 | 1.288 | 1.339 | 1.303 |
| 4 | 1.344 | 1.282 | 1.304 | 1.253 | 1.288 | 1.347 | 1.296 |
| 5 | | 1.273 | 1.296 | 1.281 | 1.286 | 1.333 | 1.302 |

**Table 4.** The dimension of each of the islets when $\theta = \frac{15°}{k}$ and $c = 0$. The modified Hausdorff dimension was used to find the dimension of these islets.

| Islet Dimension $\theta = \frac{15°}{k}$ | | | | | | | |
|---|---|---|---|---|---|---|---|
| | $k = 1$ | $k = 2$ | $k = 3$ | $k = 4$ | $k = 5$ | $k = 6$ | $k = 10$ |
| Julia set | 1.415 | 1.419 | 1.420 | 1.419 | 1.410 | 1.424 | 1.415 |
| 1 | 1.483 | 1.457 | 1.446 | 1.440 | 1.462 | 1.455 | 1.532 |
| 2 | 1.483 | 1.442 | 1.442 | 1.433 | 1.522 | 1.451 | 1.517 |
| 3 | 1.480 | 1.440 | 1.441 | 1.438 | 1.423 | 1.452 | 1.442 |
| 4 | 1.480 | 1.425 | 1.432 | 1.390 | 1.518 | 1.459 | 1.436 |
| 5 | 1.478 | 1.413 | 1.445 | 1.438 | 1.421 | 1.451 | 1.437 |
| 6 | 1.471 | 1.419 | 1.428 | 1.408 | 1.443 | 1.432 | 1.427 |

## 6. Shifted Julia Sets

The shifted Julia sets coming from the parameter subspace $(0, c)$ is now considered. Offsetting by some complex number $c$ breaks the k-fold rotational symmetry and results in a wide range of different looking filled-in Julia sets. Of particular interest is when $\text{Im}(c) = 0$ which manifests quite distinct filled-in Julia sets when $\text{Re}(c) > 0$ compared to $\text{Re}(c) < 0$. The nature of this is illustrated in the top row of Figure 8 where the cases of $c = -0.04$ (left panel), $c = 0.04$ (middle panel) and $c = 0.20$ (right panel). In all three cases the mirror symmetry about the real axis in preserved. Consistent with the Mandelbrot set (Figure 3), the origin is in the filled-in Julia set for all negative values real values of $c$. Conversely, for all positive values of $c$, the origin is not a member of the filled-in Julia set.

Also of interest is the when $\mathrm{Re}(c) = 0$ (bottom row of Figure 8). Since the imaginary axis is not colinear with a symmetry angle for this particular case of $k = 3$, both the rotational and mirror symmetries are broken. The cases of $c = -0.04i$ (left), $c = -0.12i$ (middle) and $c = -0.20i$ are shown (the fractal graphs would be mirror images for the conjugate of these values). As the magnitude of $c$ increases, a penetrating vertical cleft begins to partition the filled-in Julia set about the imaginary axis. Interestingly, the right side of the cleft is stable; maintaining its rough shape. However, the left side of the cleft is unstable and disintegrates with increasing $|c|$.

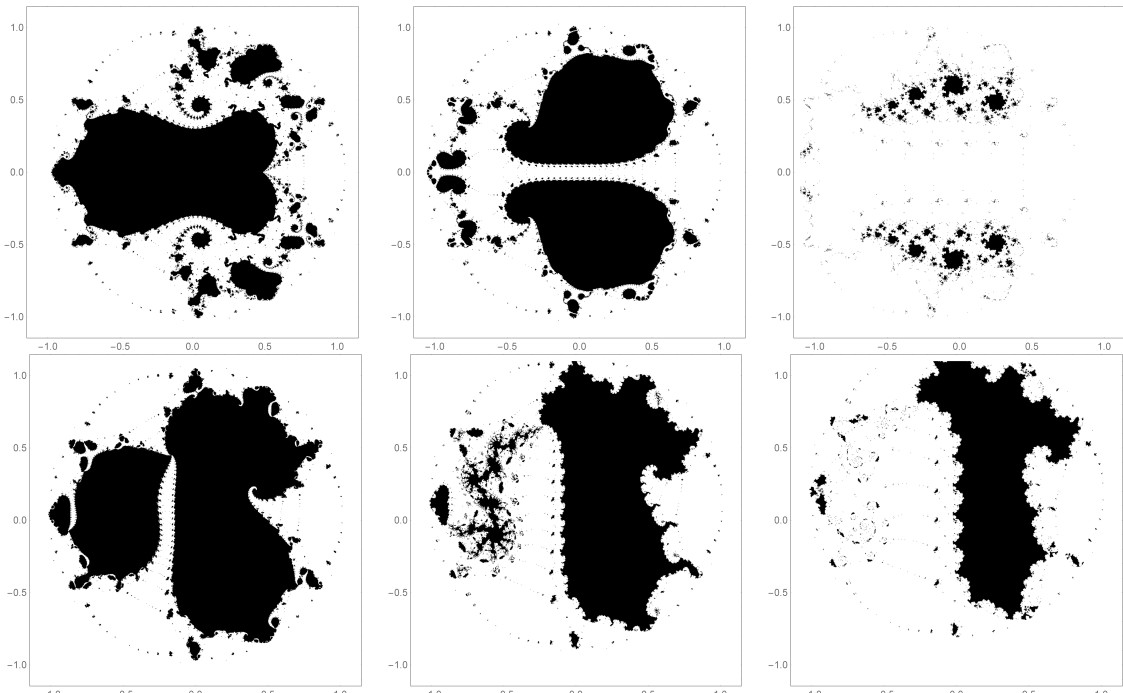

**Figure 8.** Filled-in Julia sets of $k = 3$ after an offset parameter has been applied to them. An offset of $c = -0.04$, $0.04$ and $0.20$ (top left, center and right respectively) is applied the base filled-in Julia set along the real abscissa. The bottom row shows an offset of $c = -0.04i$, $-0.12i$, $-0.20i$ (left, center and right respectively) along the imaginary ordinate. These offset, filled-in Julia sets make up the corresponding Mandelbrot set. The graphs above were constructed using $N = 8$ and $j = 32$ with a resolution of 1.5 million pixels.

The calculated box dimension of the Julia sets for 400 by 400 lattice points between $x$ and $y$ values of $-1$ and $1$ results in a beautiful, elaborate image which is displayed in Figure 9 for the case of $k = 3$. Typically, the values of the dimension (when calculable) range from approximately 1.1 to 1.8. Narrow paths of high end values of dimension cut through the image in circular arcs. It is known that dimensions of Julia sets of $f = z^2 + c$, where $c$ has values near the boundary of the Mandelbrot set, equals 2 [34]. In this work, the dimensions near the boundary of the Mandelbrot set are believed to be much less reliable than hose within it. For a more rigorous discussion of dimension at the boundary of the Mandelbrot set, see the work of Shishikura [34].

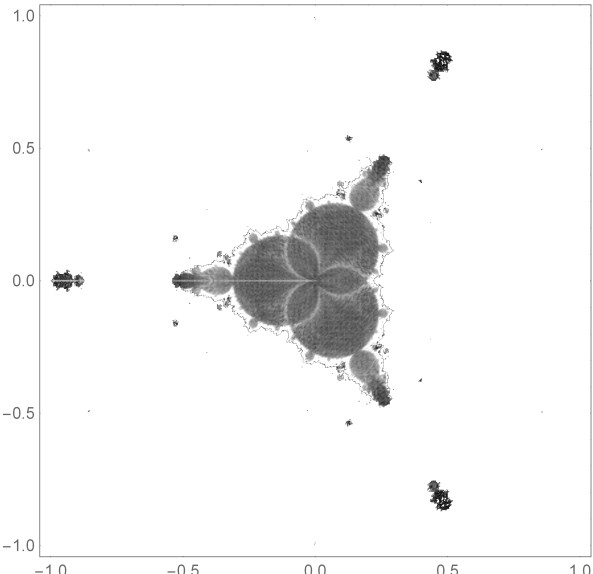

**Figure 9.** The calculated box dimension of the Julia sets for 400 by 400 lattice points in the window shown. Typical, the values of the dimension (when calculable) range from approximately 1.1 to 1.8. Low values are dark, high values are light and the white background field are areas where the Julia set is too weak to determine a dimension. It is believed that the 3-fold symmetry holds and thus the light line along the negative real axis is an artifact of the way the lattice points aligned with the Julia set itself. Further, the black outline surrounding the main body is believed to be an artifact as well. This is due to the numerical instability of the dimension calculations for these very weak Julia sets.

## 7. Centered Polygonal Functions as an Iterative Map

Shifting to a different, albeit related, point-of-few away from Julia sets and towards viewing the iterations of the centered polygonal lacunary function. Viewing these iterations as a discrete dynamical map exposes some additional features of these complicated systems. This is represented visually in two ways in Figure 10. the bottom row of graphs shows the vector field associated with the map. This visualization is good for getting an overall feel for the mapping but it is a bit difficult to identify the fixed points. The top row of graphs shows the angle of the vectors from the associated vector field. Fixed points are clearly visible in the angle plot as as they are locally encircled by rapidly changing gray scale shades.

These fixed points have interesting dependence on both phase rotation and $k$ value. Inspection of Figure 10 shows that there is, of course, always a fixed point at the origin (referred to here as the zero fixed point) and numerous fixed points near the natural boundary (referred to collectively as secondary fixed points). However, when the phase rotation $\theta = 0°$ is applied, additional prominent fixed points (primary fixed points) emerge in the vector space. Unfortunately, the position of the primary fixed points must be determined numerically but this is a rapid calculation.

One can plot the distance of the primary fixed points from the origin as $\theta$ changes. As seen in Figure 11, the points get exponentially farther away from the origin as $\theta$ increases. The degree of this movement is largely dependent on $k$ as is seen by the $k = 1$ example in Figure 11 compared to the $k = 6$ example. The primary fixed points of the $k = 1$ example take a relatively longer amount of time to start moving but move dramatically in a relatively short $\theta$ range. The inset plot of Figure 11 shows this fact, as $k = 1$ requires a larger exponential fit parameter than the subsequent $k$ values. This again shows the dependence on $k$ of the primary fixed points.

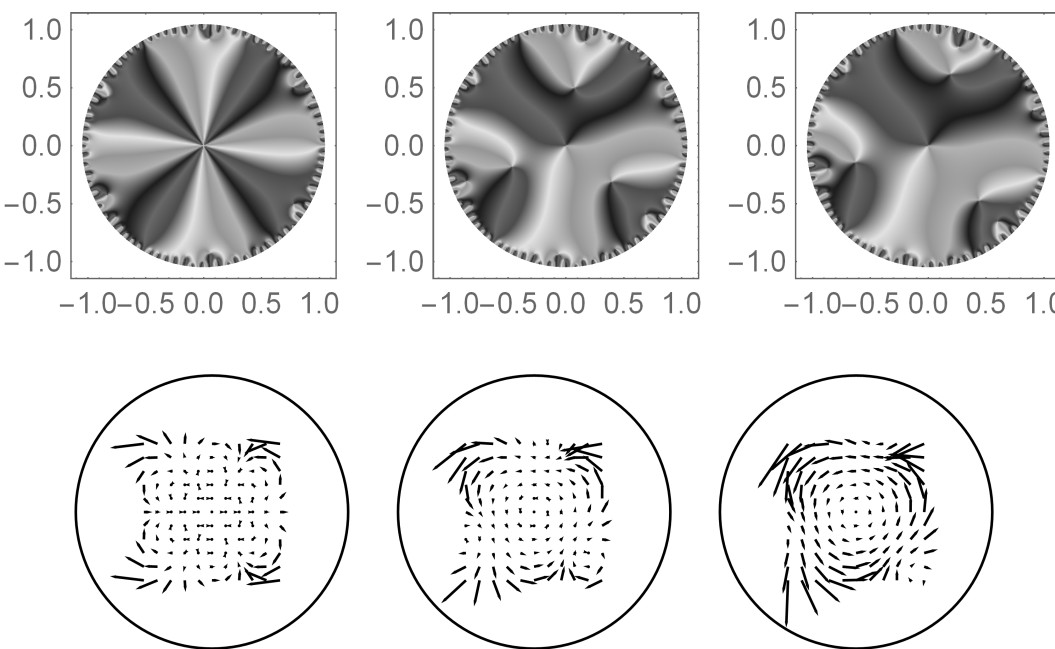

**Figure 10.** Two separate representations of the vectors fields for the discrete iterative map $f_{20}^{(3)}(e^{i\theta}z)$
The gray-scale representation (top) uses different shades to show which direction arrows are pointing.
The vector arrow representation (bottom) uses arrows to show the direction a point moves under one
iteration of the map. The phase rotation is $\theta = 0°$ for for the left column, $\theta = 9°$ for the middle column
and $\theta = 18°$ for the right column. One notices the emergence of 3 primary fixed points (see text),
when $\theta = 0°$. These fixed points move towards the natural boundary as $\theta$ increases.

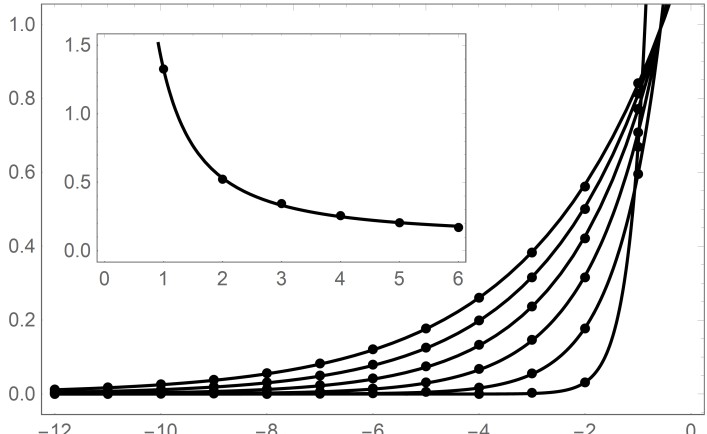

**Figure 11.** The primary plot is the radial distance of the fixed points as a function of $\theta$. Pictured are six
exponential line plots corresponding to $k$ values 1–6, with $k = 1$ being the right most curve. The value
of $k$ increase as the curves sequentially move leftward. The ordinate displays the distance from the
origin the fixed point is located (assigned variable $d$). The abscissa displays $\theta = 10^x$. The points on the
primary plot can be fit with the function $d = A10^{ax}$. The inset plot displays the exponent fit parameter $a$
(ordinate) as a function of $k$ (abscissa). These points are fit with a line from the equation $a = \frac{1.230}{k^{3/2}} + 0.095$.
Primary plot fit parameters $(k, a, A)$: $(1, 1.325, 14.172)$, $(2, 0.521, 1.978)$, $(3, 0.344, 1.561)$, $(4, 0.256, 1.389)$,
$(5, 0.204, 1.295)$, $(6, 0.169, 1.237)$.

As a visual representation, one can plot the movement of these fixed points in the complex plane
as $\theta$ changes. Figure 12 shows the way in which the fixed points move, as described above, for the case
for $k = 1$–6. Precisely $k$ primary fixed points "emerge" from the zero fixed point as $\theta$ becomes non-zero.

These fixed points arc away from the origin as $\theta$ is increased. The $k$-fold rotational symmetry is seen in those graphs.

One particularly interesting phenomenon is observed. There are precisely $k$ secondary fixed points that are nearest the the origin. As $\theta$ increases, these fixed points move inward on an apparent collision-course with the arcing primary fixed points. However, the trajectories divert course before coalescence of the fixed points is achieved. This phenomenon is neither understood in an analytic nor intuitive way by the authors, other than to note the strong similarity to the non-crossing rule for eigenvalues in elementary quantum mechanics.

Numerical analysis shows that the minimum distance between the primary and secondary fixed points decreases with increasing $k$ (visually evident in Figure 12). Perhaps more interesting is the fact that phase rotation angle where the phenomenon occurs is a very weak function of $k$. These facts are illustrated in Figure 13.

Investigating the orbit trajectories of initial points under the centered polygonal lacunary function iterative map reveals a strong sensitivity to initial conditions. Such is the case for the phase rotated, $k = 3$ iterative map example shown in Figure 14. The phase rotation is $\theta = 11.225°$ and the truncation is set to $N = 10$. One initial point considered is $z_0 = 0.15 + 0.64i$ which settles into a stable orbit after approximately 50 iterations. The nearby initial points at $z_0 = 0.15 + 0.63i$ and $z_0 = 0.15 + 0.65i$ diverge from the unit disk after 14 and 16 iterations respectively.

The trajectories are also sensitive to the truncation value, $N$. Figure 14 also compares the case where $z_0 = 0.15 + 0.64i$ but $N = 9$ instead of 10. While this point still settles on a stable orbit after approximately 70 iterative steps, that stable orbit is different than the one settled on for the case where $N = 10$. Furthermore, the trajectory for the $N = 9$ case takes a momentary excursion to the vicinity of one of the primary fixed points. This is not the done in the $N = 10$ case.

In general, the fixed points act as meta-stable attractors. Some trajectories can be found that orbit them many times before moving either to a stable orbit around the zero fixed point or diverging from the unit disk. For a very recent analysis of orbits and escape time algorithms for fractals associated with finite polynomials, the reader is directed to the work of Nazeer and Kang's group [28–31].

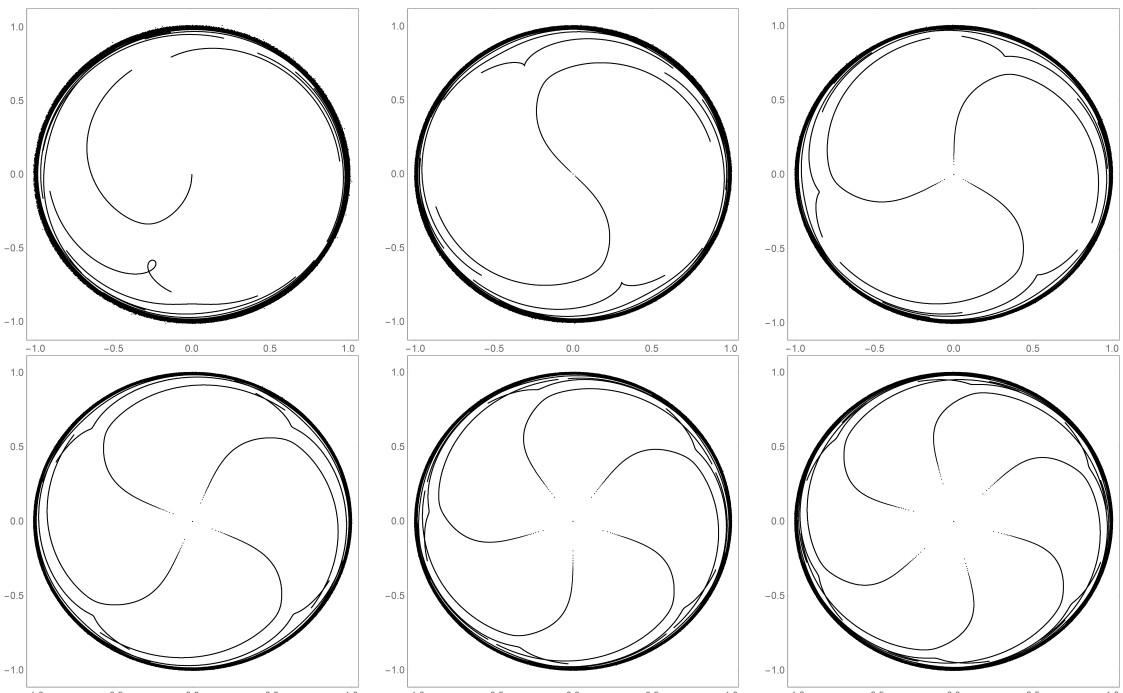

**Figure 12.** The movement of the fixed points as $\theta$ runs from $0°$ to $90°$ for $k$ values 1–6, $N = 20$. The points originating close to the center are the primary fixed points, whereas the points very close to the unit circle are the secondary fixed points.

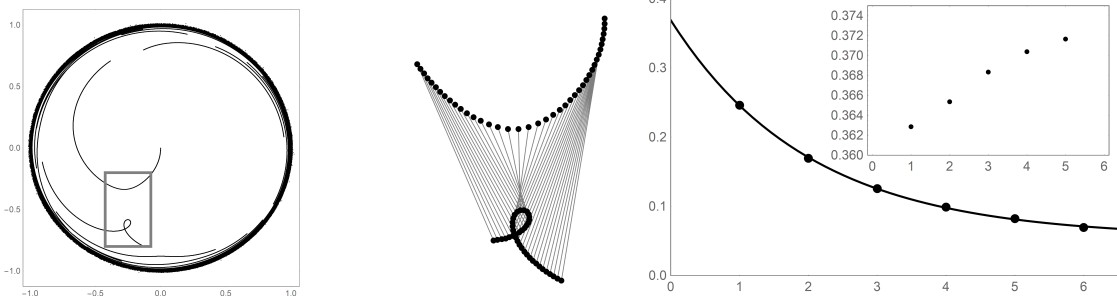

**Figure 13.** The path of the primary and secondary fixed points as $\theta$ increases (left). The middle panel is a blow up of the boxed area of the left graph . The gray tie-lines show the distance between this primary fixed point and this secondary fixed point for a given $\theta$ value. The minimum distance decreases with $k$ and seen in the right graph of Figure 13 (abscissa: k, ordinate: distance). Interestingly the $\theta$ value that gives the minimum $\theta_{min}$, is a very weak function of $k$, as shown in the inset plot (abscissa: $k$, ordinate: $\theta_{min}$).

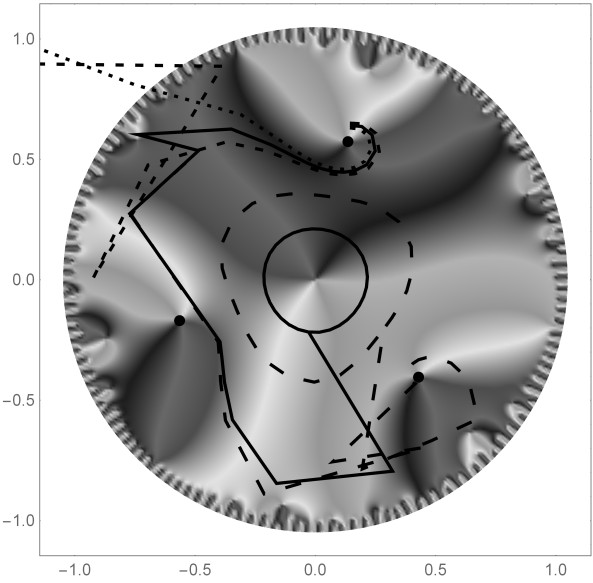

**Figure 14.** Iterative trajectories for several nearby initial points for the case of $k = 3$ and $\theta = 11.225°$. The solid line is for the initial point $z_0 = 0.15 + 0.64i$, $N = 10$ and 500 iterative steps (the stable orbit is reached after approximately 50 steps). The small and medium sized dashed lines show the trajectories for $z_0 = 0.15 + 0.63i$ and $z_0 = 0.15 + 0.65i$ respectively. Both of these trajectories exit the unit disk and diverge after 14 and 16 steps respectively. The large dashed line is again for $z_0 = 0.15 + 0.64i$ but now $N = 9$ (still 500 iterative steps). A different stable orbit is reached after approximately 70 steps and after an excursion to the area near the lower right primary fixed point.

## 8. Methods

The investigations in this work were assisted by the use of home-written Mathematica code. The programming style was largely functional rather than procedural. All computations were performed on Mathematica version 11.3.0.0. The computers used to run this program have a Intel(R) Core(TM) i5-7500 CPU @ 3.40 GHz processor (Dell: Round Rock, TX, USA), 16 GB RAM and a 64-bit Operating System. The computers also have four available cores to run four Mathematica kernels simultaneously. Large data sets were calculated by distributing calculations over up to 30 computers. This greatly reduced the time taken to calculate data as it was common for a data file to take an hour or more to be created.

The Mathematica code for generating the lacuanary functions themselves is relatively simple:

$$GF[z\_,g\_,p1\_,p2\_] := Sum[z\^g,\{n,p1,p2\}].$$

Filled-in Julia sets were constructed via the Mathematica code:

```
JSplotall[m_,j_,s_] := Module[{f}, f[y_] := GF[y, (3n^2 - 3n + 2)/2,1,m];
ListDensityPlot[Flatten[Table[Table[{x,y,If[Abs[Nest[f, z, j]] > 2,1,0]},
            {x, -1.1, 1.1, s}], {y, -1.1, 1.1, s}], 1],
                ColorFunction -> GrayLevel] // Quiet],
```

where *m* is playing the role of *N* (since *N* is pre-dedicated in Mathematica), *j* is the number of iterations and *s* is the sampling step size. The Mandelbrot sets were constructed similarly but now recording the fate of $z_0 = 0$ upon iteration as a function of offset parameter *c*.

The box dimension was calculated using the Mathematica Code:

```
JSdim[js_] := Module[dj,tj,ej,aj, tj = Transpose[js][[3]];
 dj = Apply[Plus,Table[If[tj[[j]] == 1 && tj[[j + 1]] == 0,1,0],
{j,1,Length[tj] - 1}]]; ej = Table[tj[[i]], {i,1,Length[tj],2}];
 aj = Apply[Plus,Table[If[ej[[j]] == 1 && ej[[j + 1]] == 0,1,0],
        {j,1,Length[ej] - 1}]]; N[Log[2, dj/(aj/2)]]]
```

and the modified Hausdorff dimension was calculated via the more elaborate set of functions,

```
JSxy[JS_] := Transpose[Drop[Transpose[JS],-1]]
```

```
JSonly[JS_] := Module[tt, tt = JSmaker[JS]; Drop[Union[Table[If[tt[[j]][[3]] ==
                0,tt[[j]],Null], {j,1,Length[tt]}]],1]]
```

```
pointsin[l_,rf_] := Position[Table[If[l[[i]] <= rf,0,1],
                {i,1,Length[l]}],1][[1]][[1]]
```

```
JShausDim[JS_,r_,m_,s_,bd_] := Module[th,ttt,dat, th = JSxy[JSonly[JS]]; ttt =
reldif[th]; dat = Table[If[pointsin[ttt[[j]], m r] > 20, Fit[Table[Log[n], Log[-1 +
        pointsin[ttt[[j]],n r]], {n,3,m}], {1,x},x][[2]] /.  x -> 1, bd],
            {j,1,Length[th],s}]; {Mean[dat], StandardDeviation[dat]}]
```

While there is not an overwhelming reason to choose the modified Hausdorff dimension over the standard Hausdorff dimension, it was done for two reasons. First, it is helpful to the authors for this and future studies in that it automatically provides a reasonable measure of the local dimension of a Julia set or Mandelbrot set. Second it is a better match for the programming skills of the author.

As mentioned earlier, the modified Hausdorff dimension employed throughout the work took about 4 or 5 h to complete. The timing of these two calculations were performed for primarily $N = 30$, $j = 50$ and an image resolution of 1.5 million pixels (or data points in the case of text files). However, computational time is largely dependent on *N* and *j* values as well as pixel count. The authors found that the primarily used values offered the best compromise between image quality and computation time.

## 9. Conclusions

Lacunary functions are a particularly important class of functions that exhibit a natural boundary. Even though this property causes the function to be non-analytic, lacunary functions still find many uses in physics as well as probability theory. The centered polygonal lacunary functions belong to a particular family of lacunary functions with unique properties that discern them from other lacunary functions. They show true rotational symmetry and exhibit fractal-like characteristics. This paper has shown the features of these lacunary sequences and explored much of the beautiful fractal-like nature of these sequences.

In particular, Julia sets arising from a parameter space that included a phase rotation component and an offset shift component. The quantitative feature of the Julia sets, including dimension, area and radius, were calculated. Qualitative features such as stable versus unstable islands and the structure of archipelagos were presented.

In addition to Julia and Mandelbrot sets, related iterative dynamical maps were discussed. The nature of their fixed points and dynamic trajectories were expounded on.

It is hoped that this work will provide foundation for deeper studies into the fractal characteristics of centered polygonal lacunary functions. Specific areas of follow-up could include a more thorough investigation of the phase rotation and offset shift parameter space and a detailed study of the nature of dynamical trajectories and their dependence on the various parameters of the centered polygonal lacunary functions.

**Author Contributions:** K.S. and D.J.U. conceived the investigation; L.K.M., T.V., K.S. and D.J.U. designed the investigation. L.K.M., T.V. and D.J.U. provided background for the investigation; D.J.U. wrote the Mathematica code to perform the investigation; D.R. developed distributed computing capabilities; L.K.M., T.V., K.S., D.R. and D.J.U. analyzed the data; L.K.M., T.V. and D.J.U. wrote the original draft of manuscript; L.K.M., T.V., K.S., D.R. and D.J.U. edited the manuscript.

**Funding:** This research was funded by the Concordia College Chemistry Endowment Fund and the Office of Undergraduate Research.

**Acknowledgments:** We thank Douglas R. Anderson for valuable conversation. We thank Tony Pietrzak, Thomas Holmgren, Eric Alvarez, and James Jehlik for valuable assistance with distributed computing of large data sets.

**Conflicts of Interest:** The authors declare no conflict of interest. The funders had no role in the design of the study; in the collection, analyses, or interpretation of data; in the writing of the manuscript, or in the decision to publish the results.

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
