# Peer review of "Exploration of Filled-In Julia Sets Arising from Centered Polygonal Lacunary Functions"

_fractalfract, doi:10.3390/fractalfract3030042_

Round 1

Reviewer 1 Report

This is a nice and original contribution to the literatures on Julia sets as well as lacunary functions.  The figures are beautiful!

The introduction gives a well-thought-out account of lacunary functions and their applications.  The first explanation of what a 'centered polygonal number' is comes at the start of Section 2.1.  I suggest to give a sentence (like the first sentence of 2.1) earlier on what a centered polygonal number since that term is central to the paper.

Some more explanation on why the modified Hausdorff dimension was used instead of the Hausdorff dimension would be appropriate.  On line 233, does "Though it has proven to be more accurate..." refer to the modified Hausdorff dimension's being more accurate than the Hausdorff dimension?  Is the modified Hausdorff dimension your invention in this paper, or is there a reference to it?  

Lines 22-23 state that

Hadamard’s gap theorem states that if the gaps in the powers increase such that the gap tends

23  to infinity as → , then the function will exhibit a natural boundary.

Are there other conditions relevant to the statement 

satisfying the conditions of Hadamard’s Gap Theorem [4].

on line 64?

The writing and typesetting are good, although there are some typographical errors (examples include 'require'--> 'requires" on line 126 and 'an computation time' on line 365. 

Author Response

Please see attachment. Response to both referees included.

Reviewer 2 Report

1) motivation of results should be presented main results.

2) methodology section is missing

3) there should be a section named main results and remaining section should be as subsection.

4) Write algorithm required for construction of Julia and mandelbrot sets.

5) there are many papers in literature for example Y". Cho, A. A. Shahid, W. Nazeer, and S. M. Kang, “Fixed point results for fractal generation in noor orbit and s-convexity,” SpringerPlus, vol. 5,no. 1, p. 1843, 2016.""Y. C. Kwun, M. Tanveer, W. Nazeer, K. Gdawiec, and S. M. Kang,

“Mandelbrot and julia sets via jungck–cr iteration with s–convexity,” IEEE Access, 2019.", 10.30538/oms2018.0017 and references therein. Why the images are black and white. Also cite these reference to update literature review. You may also consider references therein.

6)The title talks about Julia sets, but in the whole paper you consider only filled julia sets. The julia set is boundary of filled julia set i think. So title must be revised.

Overall the paper is interested and I enjoyed reading this paper, but before publication it need revision.

Good luck

Author Response

(The authors gave the same response as above.)
